# Implications for methenamine hippurate use in recurrent urinary tract infection management: Formaldehyde resistance and altered urinary composition

**Niamh C. Hodgkinson[1,2], Tabarak Al-Rubaye[1,2], Thomas C. P. Reed[1], Catherine Mowbray[1], Daniel Sarkissian[1], Louise Cowley[1,2], Frank Sargent[1], Judith Hall[1], Priyanka Krishnaswamy[3,4], Chris Harding[4,5]\*, Phillip D. Aldridge[1]\***

1 Biosciences Institute, Faculty of Medical Sciences, Newcastle University, Newcastle upon Tyne, United Kingdom, 2 School of Biomedical Sciences, Nutrition and Sport Sciences, Faculty of Medical Sciences, Newcastle University, Newcastle upon Tyne, United Kingdom, 3 Gynaecology and Women's services, Royal Victoria Infirmary, Newcastle upon Tyne Hospitals NHS Foundation Trust, Newcastle upon Tyne, United Kingdom, 4 Translational and Clinical Research Institute, Faculty of Medical Sciences, Newcastle University, Newcastle upon Tyne, United Kingdom, 5 Urology Department, Freeman Hospital, Newcastle upon Tyne Hospitals NHS Foundation Trust, Newcastle upon Tyne, United Kingdom

\* c.harding@nhs.net (CH); phillip.aldridge@ncl.ac.uk (PDA)

## Abstract

Methenamine is a urinary antiseptic used to prevent urinary tract infections (UTI) via conversion to formaldehyde in the urinary tract. Methenamine hippurate (MH) is non-inferior compared to antibiotic (ABX) prophylaxis to manage recurrent UTI (rUTI) as demonstrated in the clinical trial ALTAR. Treatments such as MH, can improve antibiotic stewardship, as the primary treatment option for UTI is antibiotics. However, MH exhibits an elevated incidence risk with respect to breakthrough UTI as defined during ALTAR. Formaldehyde is highly toxic, while also a common byproduct of cellular metabolism. Powerful detoxification pathways exist to overcome formaldehyde toxicity. One example is the thiol-dependent metabolism of formaldehyde to formate in bacteria. The urinalysis of ALTAR urines detected formaldehyde in 85% of participant urines who were taking MH. HPLC analysis of a subset of urines from MH and ABX ALTAR participants, determined a significant change in urine composition. This included elevated levels of formate in urines from MH users. The thiol-dependant formaldehyde detoxification system of *Escherichia coli* is encoded by the *frmRAB* operon. The genes *frmAB* encode the enzymes responsible for detoxification, while *frmR* encodes a repressor of the system. ALTAR derived *E. coli* isolates were screened for growth in the presence of formaldehyde with 5.8% able to grow in > 1 mM formaldehyde. Bioinformatics identified 4 *frmR* alleles encoding non-functional FrmR variants and two plasmid-encoded *frmA* homologues. Growth in artificial urine confirmed that *E. coli* was susceptible to methenamine-formaldehyde conversion at pH6.0 and 5.6. All strains encoding *frmR* alleles grew in the presence of > 1 mg/

**Data availability statement:** All relevant data are within the manuscript and its Supporting information files. Genomic sequence data used for this study is available: PRJEB85317 (https://www.ebi.ac.uk/ena/browser/home).

**Funding:** The funders had no role in study design, data collection and analysis, decision to publish, or preparation of the manuscript. The ALTAR trial was funded by NIHR HTA 13/88/21 to CH and part funded this study. CM received a salary for their contribution to this project from NIHR HTA 13/88/21. A Newcastle University Wellcome Trust Translational Partnership award (ref: NU015900) awarded to PA, JH, FS and CH provided financial support for the HPLC analysis. The contributions of NH and TA were made during their MSci Biomedical Sciences and MRes research projects, respectively, supported by Newcastle University. TR and FS are funded by BBSRC grant BB/Y004302/1, that included the salary for TR. The salary of DS was supported by funding from a 'The Urology Foundation innovation and Research Award 2023' grant awarded to PK, JH, CH and PA (ref: NU015458). LC was supported by a Microbiology Society Harry Smith Vacation Studentship (GA004815) for her contribution to this project awarded to LC and PA.

**Competing interests:** The authors have declared that no competing interests exist.

ml methenamine at pH 5.6. The identification of FDH$^R$ in a clinical context and the changes in urine composition can improve the managed use of MH. However, a mindset change is needed to accept that MH, like antibiotics, has its own associated risks, including bacterial resistance.

## Author summary

The mainstay of treatment for urinary tract infections is short course antibiotics, often sufficient to clear the infection. However, a significant proportion of UTIs are recurrent, leading to frequent antibiotic consumption. The drive for better awareness of antibiotic prescription and antibiotic stewardship initiatives has led to research into alternatives to antibiotics for both management and prevention. The clinical trial ALTAR has determined that an alternative to antibiotic prophylaxis, methenamine hippurate, is not inferior with respect to improving the prevention of recurrent UTI episodes. Methenamine hippurate acts by releasing formaldehyde into urine through the breakdown of methenamine in acidic conditions generated by hippurate. The assumed mode of action is therefore that formaldehyde acts as a urinary antiseptic, and that as it is not an antibiotic, resistance cannot develop. Our data argues that uro-associated Escherichia coli is adapting to exposure to formaldehyde leading to methenamine hippurate resistance. Methenamine hippurate has its merits to improve antibiotic stewardship when treating recurrent UTI patients. However, a mindset change is needed by healthcare practitioners and patients to accept that, like antibiotics, there are associated risks, including bacterial resistance.

## Introduction

Urinary tract infections (UTI) are predominantly a bacterial infection, and the associated symptoms are the result of an inflammatory response to pathogens in the bladder (cystitis) or kidneys (pyelonephritis). UTI is one of the most common reasons for antibiotic prescription [1]. For the majority of acute UTI cases, a short course (3–5 days) of antibiotics is sufficient to clear the infection. However, in up to 50% of cases recurrence is observed, leading to further antibiotic treatment [2]. Recurrent UTI patients will often be recommended to try prophylactic treatment usually as an extended-course (months) of low-dose prophylactic antibiotics [3]. The link between antibiotic use and the development of antimicrobial resistance in uro-associated bacterial species or the acquisition of multi-drug resistant uro-associated bacteria is not in question [4].

Antimicrobial resistance (AMR) is a leading threat to the global healthcare sector. A key driver within healthcare to combat AMR is the implementation of antibiotic stewardship [5]. However, when the primary treatment option is antibiotic use, as is the case for UTI, the balance between adequate versus judicious treatment can be

difficult. For example, raising awareness with respect to correct diagnosis for healthcare providers may reduce antibiotic prescription [6].

Alongside antibiotic stewardship initiatives there is renewed interest in seeking alternative non-antimicrobial treatment strategies for UTI. Examples include the use of D-mannose, probiotics, and several candidate vaccines [7,8]. These alternatives have the potential to result in a reduction in UTI frequency and consequent antibiotic use. Recently the UK based National Institute for Health and Care Excellence (NICE) have recommended the use of the urinary antiseptic methenamine hippurate (MH) for UTI prevention in their updated guideline on antimicrobial prescribing for UTI [9]. Methenamine has been employed to treat UTI since 1894 [10].

The accepted mode of action for MH is the pH dependent cleavage of methenamine to formaldehyde and ammonia in the distal tubules of the kidney [10]. Hippurate acts as the acidifying agent [11]. Formaldehyde is assumed in this context to be a urinary antiseptic, and there is also the associated assumption that resistance will not develop [12]. However, urinary formaldehyde concentrations found in human urine after treatment with MH (~ 300–1300 µM) are at best bacteriostatic [12].

Our bodies and bacterial species naturally generate formaldehyde during amino acid metabolism [13]. In fact, formaldehyde can be detected in our blood in the range of 20–100 µM, while bacterial methyltrophs can have, at times, an internal formaldehyde concentration of up to 1000 µM during growth on methanol [14,15]. The toxicity of formaldehyde is via its reactivity as a electrophile, rapidly reacting with thiol and amine groups of proteins and DNA, generating both reversible and irreversible crosslinks [13]. Due to its toxicity, formaldehyde is frequently used as a disinfectant during sterilisation. However, many organisms have pathways involved in detoxifying formaldehyde in part due to its important in cellular metabolism. In bacteria three detxofication pathways have been described: thiol-dependent, pterin-dependent and sugar phosphate dependent pathways [13,15]. The most common of these three is the thiol-dependent pathway. For *Escherichia coli* (the focus of this study)*,* thiol-dependent formaldehyde detoxification is driven by expression of the *frmRAB* operon [16]. On entering a bacterial cell, formaldehyde spontaneously interacts with reduced glutathione, generating *S*-hydroxymethylglutathione (HMGS). FrmA catalyses the oxidation of HMGS, in a NAD+ dependent manner, to form *S*-formylglutathione and NADH. *S*-formylglutathione is subsequently hydrolysed by FrmB generating formate and reduced glutathione. FrmR is a negative transcriptional regulator of *frmRAB* expression that directly interacts with formaldehyde in the cell [17].

The ALTAR study determined that MH treatment was non-inferior to prophylactic antibiotics in terms of UTI prevention [11]. MH treatment negatively impacted AMR carriage in the gut reservoir and reduced antibiotic consumption. ALTAR did highlight limitations of MH with a marginally higher rate of UTI episodes in those taking MH compared to the prophylactic antibiotic group (1.38 versus 0.89 UTI episodes per year) [11]. The absolute difference in UTIs, 0.49 episodes per year, was considered of limited clinical significance, while being biologically significant with respect to changes in the carriage of AMR in the gut. Our objectives in this study were, therefore, to understand what factors could contribute to this elevated risk of a UTI during MH use compared to daily low dose antibiotics.

## Results

The study protocol of ALTAR compared MH treatment to daily low dose antibiotics (ABX) in the female (> 18 yrs old) population, where the antibiotics in use were nitrofurantoin, trimethoprim or cefalexin [11,18]. Demographically the participants were well balanced between each arm with respect to age (51.1 ± 18.3 years), weight (72.5 ± 16.6 kg) and menopausal status (40% pre- and 60% peri/post-) [11]. One hundred and 95 participants (MH = 98; ABX = 97) completed the trial that included a 12-month treatment period and a 6-month post-treatment follow-up. ALTAR participants provided regular midstream clean-catch urine and perineal swab samples (baseline, 3, 6, 9, 12, 15 and 18 months) for analysis and the prospective isolation of *E. coli*. Participants that had provided ≥ 5 urine samples (n = 681) constituted 58% of all samples from 65 MH (66%) and 51 (52%) ABX participants. We have recently described the sequence analysis of 191 ALTAR derived

*E. coli* isolates that represented 45% of all isolates [19]. Here this secondary analysis associated with ALTAR, focusses on these 681 urines and the 191 sequenced *E. coli* isolates.

## Formaldehyde levels in urine

Methenamine in acidic urine is converted into formaldehyde and ammonia. Musher and Griffith (1974) showed that 890–1330 µM formaldehyde is generated from 1 mg/ml of methenamine in artificial urine after 3 hours when the pH was < 6.0 [12]. Evidence suggests that the concentration range found in urine is between 400 and 1500 µM but is dependent on the pH of the urine [20]. We were able to detect > 0.7 µM formaldehyde in 49% of samples from the 65 participants in the MH arm. Calculation of the average formaldehyde concentration for both ALTAR arms showed that formaldehyde was detected between 90 and 360 days (**Fig 1A**). However, MH participants show considerable variation in detectable formaldehyde (n = 186; 0.88 to 122.1 µM) (**Fig 1B**). In contrast, formaldehyde was barely detected in ABX participants (**Fig 1**: n = 56; 0.83 to 49.1 µM) Approximately 30% of MH-derived samples between days 90–360 had levels of

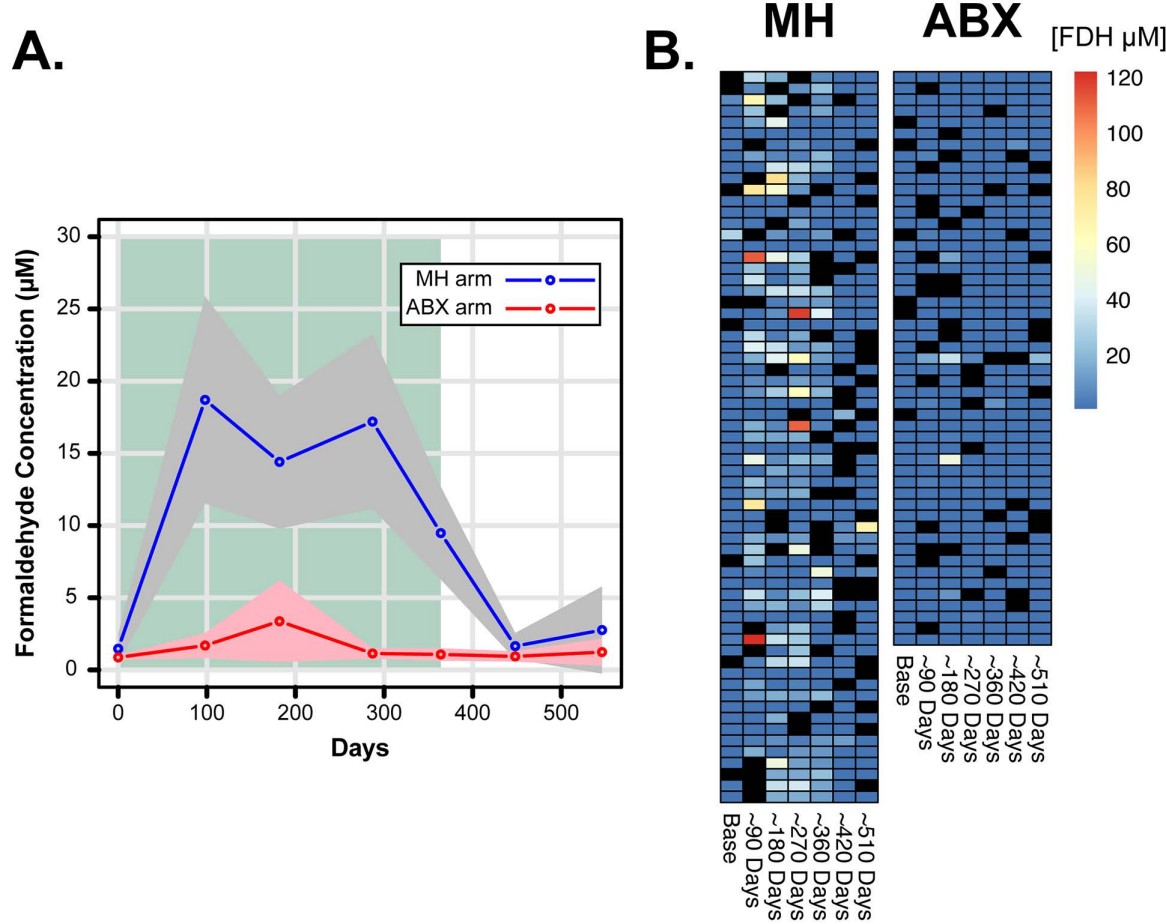

**Fig 1. Detection of formaldehyde in ALTAR urine samples. A:** Average formaldehyde detected across participants that had provided ≥ 5 samples during ALTAR. The grey and pink polygons indicate the confidence intervals calculated from the data. The green background indicates the period of treatment during ALTAR **B.** Heatmap of all data to show the variability in formaldehyde detection. Black squares are missing samples. All samples used correlate to routine 0-, 3-, 6-, 9-, 12-, 15-, and 18-month sampling and does not include UTI samples. For plotting purposes data is presented in days.

formaldehyde below our detection limit of 0.7 µM, compared to 82% at baseline. These data are consistent with previous studies of methenamine conversion, in that we detect µM levels of formaldehyde in the urine of MH users [20].

### *Escherichia coli* adaption to formaldehyde exposure

The early work of Griffith and Butler argued that the concentration of formaldehyde detected in their studies was at most bacteriostatic [12,20]. For the *E. coli* strain CFT073, often seen as a model uropathogenic *E. coli* strain [21], we determined the MIC to be ≤ 1 mM formaldehyde. This is towards the upper limits of detectable formaldehyde during MH treatment. All 191 sequenced ALTAR *E. coli* isolates were screened for growth in 1, 2, 2.5, 3 and 3.5 mM formaldehyde (**Fig 2**). As the accepted dogma argues against formaldehyde resistance, we were surprised that 11 isolates associated with 5 patients exhibited elevated formaldehyde MIC values. Isolates with an elevated formaldehyde MIC were confirmed in further growth assays defining 1 MIC as > 3 mM, 9 MIC as > 2 mM and 1 as > 1.75 mM formaldehyde (**Fig A and B in S1 Text**). This argues against the assumed dogma that formaldehyde is an antiseptic and thus immune to the development of bacterial resistance [12]. We defined a MIC > 1 mM as FDH$^R$ and a MIC ≤ 1 mM as FDH$^S$.

The phylogeny of FDH$^R$ isolates was visualised and cross referenced with participant metadata. ALTAR *E. coli* isolates demonstrate diversity across the phylogenetic clades of *E. coli* (**Fig 2**). Importantly the FDH$^R$ phenotype does not associate with a specific clade of *E. coli*, instead the data suggests that *E. coli* as a species can adapt to formaldehyde exposure. Four of the 5 participants associated with the FDH$^R$ phenotype were from the MH arm, defined as cases A, B, C, and D (**Fig 3**). The 5th participant (defined as case E) was from the ABX arm on the antibiotic nitrofurantoin, with a single FDH$^R$ baseline swab isolate (**Fig B in S1 Text**).

For the MH participants, several important observations to strengthen our argument for formaldehyde resistance were evident (**Fig 3**). *E. coli* was isolated from cases A to D a total of 17 times, for case A: 2 swab and 3 urine isolates; for case B: 3 swab and 3 urine isolates; for case C 2 urine isolates; and for case D 3 swab and 1 urine isolate (**Fig 3**). FDH$^R$ *E. coli* was identified in both the urine and swab samples: 4 times in swabs and 7 times in urine. In two of the four MH participants (cases A and C) isolation of FDH$^R$ *E. coli* from their urine coincided with low formaldehyde detection (**Fig 3A and 3C**). Consistently, in case D after approximately 200 days the absence of the FDH$^R$ isolate is mirrored by increased formaldehyde detection (**Fig 3D**). Importantly, genotyping using the Achtman multi-locus sequence typing scheme [22] and phylogenetic analysis strongly argues for *in situ* evolution of the FDH$^R$ in two participants over ~180 – 270 days (**Fig 3A** and **3B**: blue and green strains). Therefore, this suggests that over time isolates have the capacity to acquire FDH$^R$ during MH treatment.

### Genetic evidence for deregulation of *frmRAB* leads to FDH$^R$

A logical starting point for the FDH$^R$ phenotype is genetic changes impacting the output of the *frmRAB* operon and the ability for *E. coli* to detoxify formaldehyde. Using an in-house protein BLAST database of all annotated open reading frames from the ALTAR *E. coli* isolates, we found that 8 of the 11 FDH$^R$ isolates had amino acid substitutions in FrmR (**Table 1**). The other three isolates had a wild type chromosomal *frmRAB* locus but carried two unrelated plasmids encoding at least, *frmA and frmB* genes interspersed with other genes (**Fig C in S1 Text**). One of the amino acid substitutions E6stp has been previously described [23], while the others are novel substitutions. Based on phylogenetic analysis, the variants P5R and G47S have plausibly evolved from FDH$^S$ strains during MH use on ALTAR (**Fig 3A** and **3B** respectively), while V86D has co-evolved alongside E6stp from a potential common unidentified ST405 parent strain (**Fig 3C**).

Plasmid-encoded formaldehyde resistance has previously been described on isolation of the plasmid pVU3695 [24,25]. A 1.1 kb pVU3695 region encoding *adhC* (*frmA*) was available for comparative analysis [25]. The case D plasmid (**Table 1** and **Fig 3D**) exhibited 99% identity at the DNA level to the *adhC* region of pVU3695. This plasmid encodes an IncFIB replicon, in contrast pVU3695 was predicted to encode an IncL/M replicon [26]. The nucleotide sequence for the case E plasmid (**Table 1** and **Fig C in S1 Text**), an IncN plasmid, shared 88% identity to pVU3695.

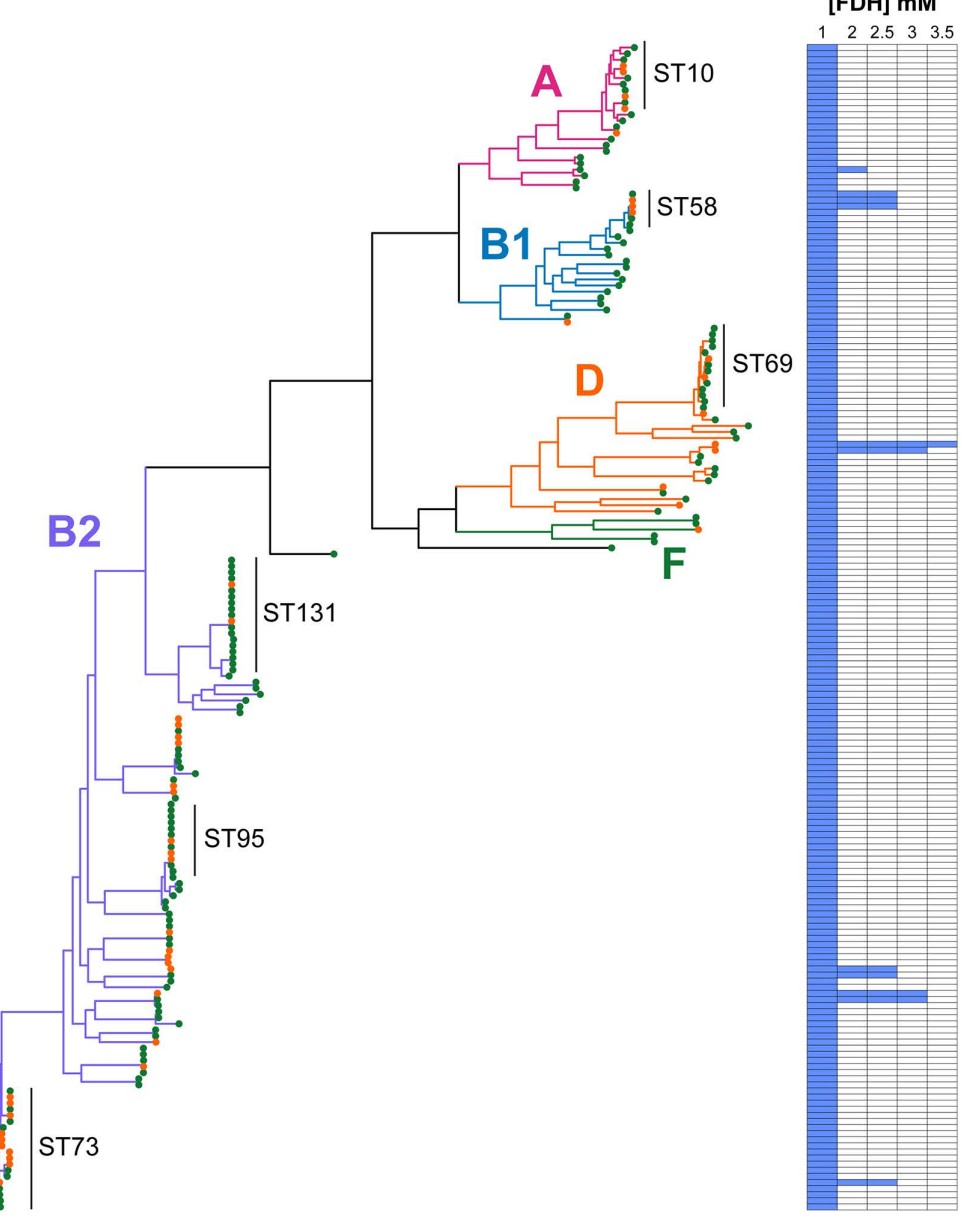

**Fig 2. Detection of formaldehyde resistance (FDH^R) in sequenced ALTAR *E. coli* isolates.** Colours of tree branches represent the major clades from *E. coli*. Common sequence types (ST) defined by the Achtman MLST scheme are shown for reference. The branch tips are coloured to represent swab (green) or urine (orange) isolates. The heatmap is a binary representation of where growth was observed at increasing concentrations of formaldehyde. All strains were screened n = 1 and any FDH^R phenotype confirmed in a minimum of n = 3 growth assays (**Fig A and B in S1 Text**).

When comparing the coding sequences of FrmA, case D FrmA and AdhC from pVU3695 shared 89% identity to the chromosomal encoded FrmA from CFT073, while sharing 99% identity to each other. In contrast FrmA from case E shared 95% identity to FrmA from CFT073 and 91% identity to AdhC from pVU3695. This data suggests that FDH^R in *E. coli* can be derived from spontaneous base substitutions in the repressor *frmR* or by horizontal transfer of plasmids encoding *frmAB* homologues.

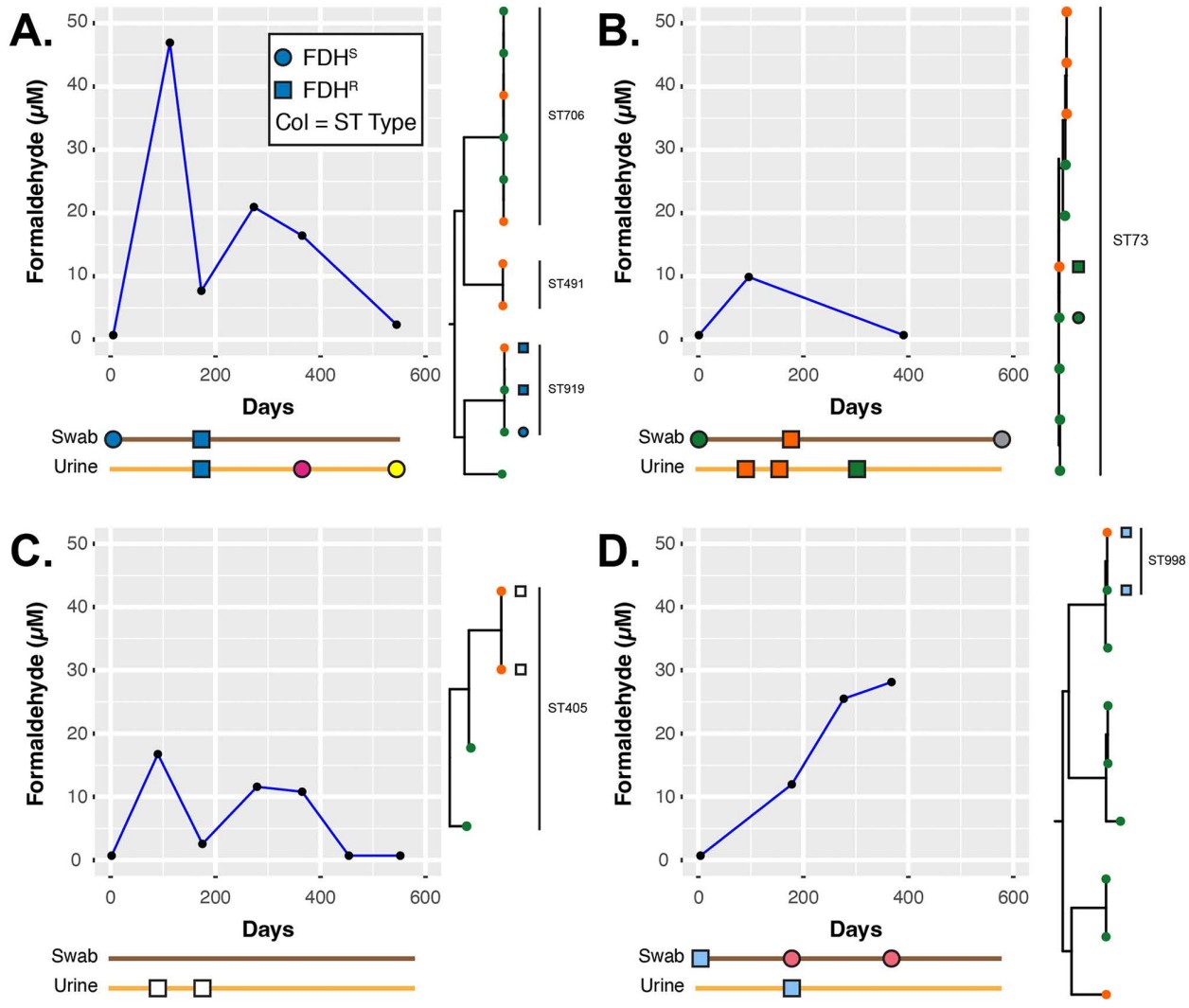

**Fig 3. Case profiles of the four MH participants (defined here as A., B., C. and D.) identified to carry FDH^R _E. coli_ isolates.** The schematic below each graph represents the timeline based on the x-axis of the plot for when _E. coli_ was isolated, with the shape defining the FDH phenotype (See key shown in A.). The colour of the objects, in the timeline schematics, reflects the isolates genotype based on the Achtman MLST scheme. Next to each plot is a clade view, taken from the phylogenetic tree in Fig 2, to show the phylogenetic relatedness of specific isolates, the same shape and coloured objects, as used in the timelines, are included in these trees to indicate the specific isolates. The line graphs show the detected formaldehyde from the urine samples, in A. & C. reduced formaldehyde detection coincided with the identification of FDH^R _E. coli_ isolates.

## _frmR_ mutants deregulate the _frmRAB_ operon

Denby et al (2016) described the deregulation of the _frmRAB_ operon by E6stp (Fig 3B: orange squares). To confirm that P5R, G47S and V86D were responsible for the FDH^R phenotypes observed, we transferred these variants using a CRISPR/Cas strategy to CFT073 [27,28]. All three mutations led to increased MIC sustaining growth in up to 3 mM formaldehyde (Fig 4A). Gene expression analysis of _frmA_ and _frmB_ relative to CFT073 in the absence of formaldehyde confirmed the deregulation of the _frmRAB_ operon (Fig 4B). Furthermore, when challenged with 1 mM formaldehyde an increased detoxification rate consistent with the FDH^R phenotype was observed (Fig 4C). This data supports the hypothesis that these amino acid substitutions deregulate _frmRAB_ expression, thus generating the FDH^R phenotype.

**Table 1. FrmR variants isolated during ALTAR.**

| Case | Source | Month Isolated* | Seq. Type | FrmR | Plasmid |
|------|--------|-----------------|-----------|------|---------|
| A | Swab | 6 | 919 | P5R | |
| | Urine | 6 | 919 | P5R | |
| B | Swab | 6 | 58 | E6stp | |
| | Urine | 3 | 58 | E6stp | |
| | | 5 (UTI) | 58 | E6stp | |
| | | 10 (UTI) | 73 | G47S | |
| C | Urine | 3 | 405 | E6stp | |
| | | 6 | 405 | V86D | |
| D | Swab | 0 | 998 | WT | IncFIB |
| | Urine | 6 | 998 | WT | IncFIB |
| E | Swab | 0 | 399 | WT | IncN |

* if just a month is given, *E. coli* was isolated during regular sampling. (UTI) reflects isolates identified during a reported UTI.

## Urine composition

The ability of *E. coli* to adapt to formaldehyde exposure by deregulating its detoxification system provides one explanation for some of the risk of UTI in individuals taking MH. However, alone it is not sufficient to explain the risk associated with MH use. To complement the FDH$^R$ analysis, we tested the hypothesis that MH treatment may also alter urine composition. A carbohydrate/ metabolite HPLC analysis using 120 samples from 20 MH participants exhibiting detectable levels of formaldehyde during ALTAR was performed. As a comparator we chose 55 samples from 10 ABX participants, where we could not detect formaldehyde and metadata showed they had not switched treatment arm.

Plotting peak area (PA) against ALTAR trial arm showed elevated PA in the MH arm compared to the ABX arm (**Fig 5A**; $p < 0.0001$). This significant change equated to 15 of 49 identified components, when in contrast only 3 out of 49 exhibited higher PA detection in the ABX arm compared to MH (**Table A in S1 Text**). Three components to show elevated levels in MH urine samples were formaldehyde, formate and glucose (**Fig 5B**). The identification of formaldehyde and formate was expected due to methenamine conversion and the ability of *E. coli*, other bacterial species and our own bodies to detoxify formaldehyde to formate. The elevated glucose levels were not expected and does raise some concern in potentially generating a urine environment that would promote microbial growth with such an accessible carbon source.

## Impact of pH and Methenamine conversion

The exposure of *E. coli* to methenamine and thus formaldehyde in urine, during treatment, requires the rate of formaldehyde production to be greater than detoxification. To test this, we conducted experiments in artificial urine with an adjusted pH to mimic MH use (pH 6.5, 6.0 and 5.6). Using the CFT073 *frmR* variants and Δ*frmR* in comparison to CFT073 (*frmR*$^+$) we demonstrate that formaldehyde detoxification is not influenced by pH changes (**Fig D in S1 Text**). The *frmR*$^+$ strain retained a formaldehyde MIC of ~ 1 mM, while Δ*frmR* and the *frmR* variants all showed robust growth in up to 1.75 mM formaldehyde. *frmR*$^+$ growth was inhibited by > 0.5 mg/ml methenamine at pH 6.0 and pH 5.6 (**Fig E in S1 Text**). In contrast, Δ*frmR* and *frmR* variants only showed moderate inhibition in the presence of 1.5 mg/ml methenamine at pH 5.6 (**Fig E in S1 Text**). All strains grew well when exposed to methenamine at pH 6.5, consistent with these conditions being inefficient for formaldehyde generation.

To determine whether formaldehyde generation or detoxification were dominant, we exposed *frmR*$^+$ and Δ*frmR* to 0.25 or 0.5 mg/ml methenamine from T0 during batch culture growth and measured the formaldehyde concentration in comparison to the growth of *E. coli* (**Fig 6**). Consistent with the other growth experiments, pH 6.5 showed no growth inhibition

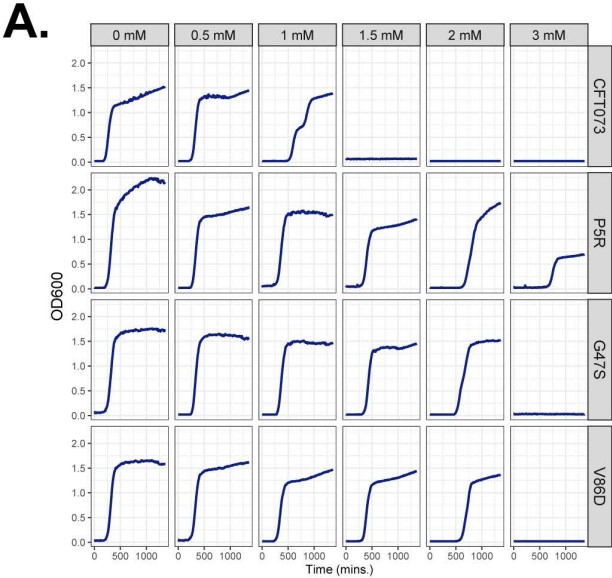

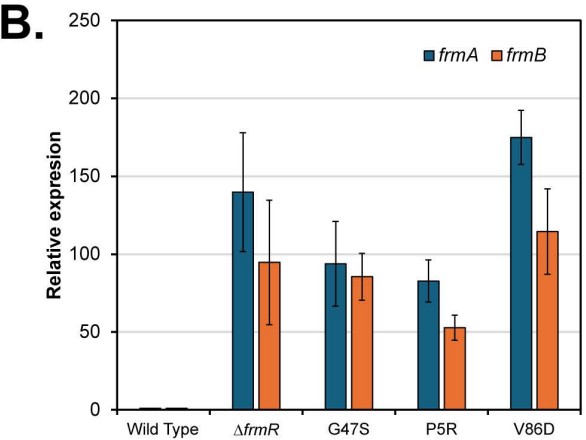

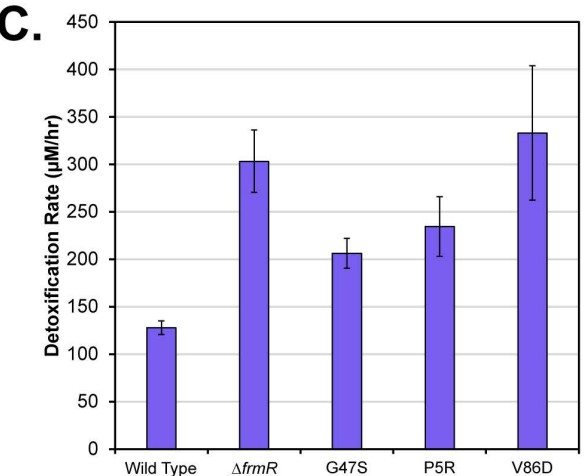

**Fig 4. Characterisation of 3 *frmR* variants introduced into CFT073 by CRISPR/Cas gene editing. A.** Growth assays of each variant compared to CFT073 in the noted concentrations of formaldehyde. **B.** Expression analysis for *frmA* and *frmB* of each variant compared to CFT073 in uninduced conditions. Included here is a scarless deletion of *frmR* (Δ*frmR*) for comparison. **C.** Detoxification rates determined during growth assay when each variant was challenged from Time 0 with 1 mM formaldehyde. All data represents a minimum of n = 3 biological independent repeats.

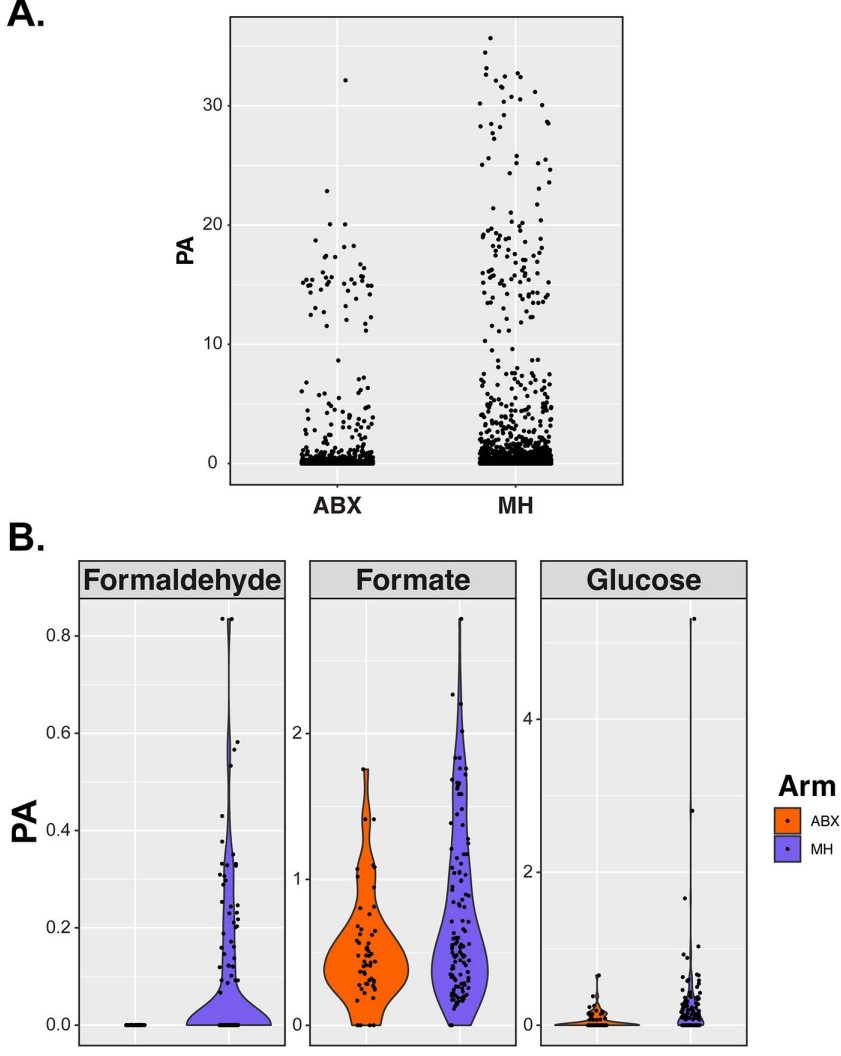

**Fig 5. Comparison of identified HPLC peak areas from analysis of 175 urine ALTAR samples. A.** scatter plot of all identified component peak areas > 0.07 demonstrating elevated levels in MH users. **B.** Component violin plots comparing known components between arms.

or formaldehyde production for either strain. A similar pattern was also observed for pH 6.0 as no change in growth was detected (**Fig 6**). However, formaldehyde was detected within the first 2 hours up to approximately 0.25 mM, levels known not to be sufficient to inhibit *E. coli* growth. In contrast, at pH 5.6 formaldehyde detection peaked at ~0.4 mM when *frmR+* was exposed to 0.5 mg/ml methenamine after 1 hour of incubation dropping down to < 0.1 mM over the next 4 hours (**Fig 6**). This initial peak was sufficient to inhibit the growth of *frmR+* in comparison to Δ*frmR*, generating a detectable lag in the growth of *frmR+* (**Fig 6**: pH 5.6 panel; p = 0.05). This was consistent with the inhibition of *frmR+* and Δ*frmR* when exposed to 1 mM formaldehyde in complex media. Growth of *frmR+* was inhibited until the formaldehyde concentration dropped below 0.4 mM generating an extended lag phase (**Fig F in S1 Text**). This suggests that if methenamine concentrations can be maintained > 0.25 mg/ml, generating > 0.3 mM formaldehyde, there is the potential to inhibit *E. coli* growth either completely or sufficiently that natural defence mechanisms, including urine voiding, can prevent *E. coli* establishing itself in the bladder, thus reducing the chance of UTI symptoms.

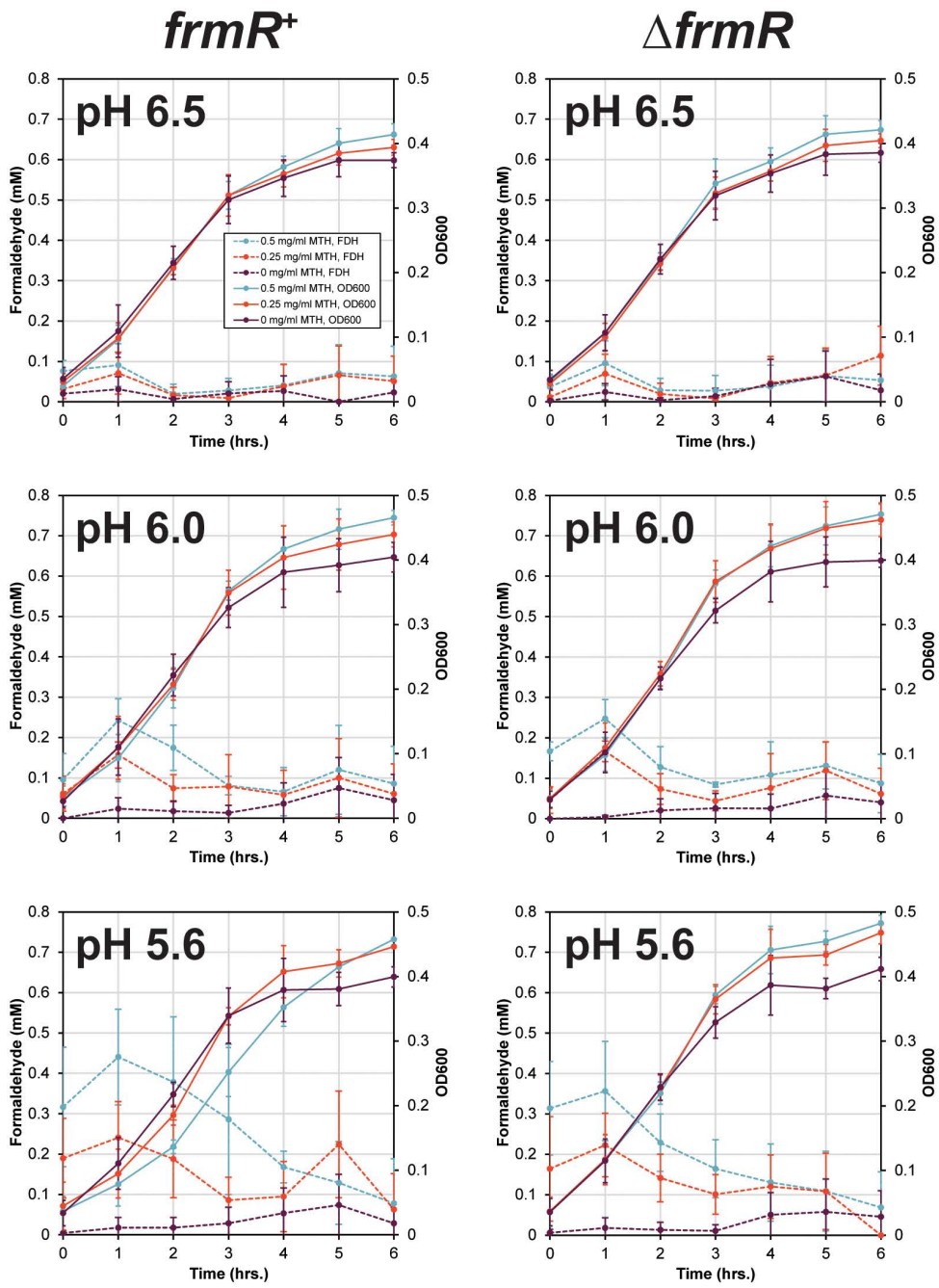

**Fig 6. Impact of pH on the growth of *frmR*⁺ and Δ*frmR* strains when exposed to methenamine in artificial urine.** Data shown is the average of 3 independent repeats for all conditions. Formaldehyde concentration detected in the cultures is plotted on the primary y-axis with culture density represented by OD600 values on the secondary y-axis. Data is colour coordinated with respect to the presence of absence of methenamine in the cultures (see key in top left graph).

## Discussion

Methenamine use has been based on the fundamental assumption that acidic conditions encountered in urine lead to the breakdown of methenamine to formaldehyde. Due to the toxic properties of formaldehyde and its classification as an antiseptic it has therefore, also been assumed that bacterial resistance cannot develop [12]. The clinical trial ALTAR provided a unique opportunity to investigate the impact of methenamine use in comparison to prophylactic antibiotics. The key finding from our secondary analysis is that *E. coli* can adapt during long-term exposure to μM concentrations of formaldehyde. This adaptation is genetically acquired and can occur via two mechanisms a) the acquisition of base substitutions in the repressor *frmR* leading to deregulation of the *frmRAB* operon; or b) the horizontal acquisition of plasmid encoded *frmAB* homologues. The outcome of either route is the detection of a formaldehyde resistance phenotype: FDH[R].

The concept of formaldehyde resistance in *E. coli* is not new, dating back to 1979–1982 when FDH[R] isolates were first described [29]. However, identifying FDH[R] during ALTAR provides direct clinical context to the potential impact. Kaulfers and colleagues provided evidence that early FDH[R] isolates were a result of horizontal transfer of plasmid encoded *frmA* [24]. Here we have identified two such cases that share similarity to the region encoding *frmA* on pVU3695. Furthermore, we provide direct evidence that *E. coli* can adapt via deregulation of the *frmRAB* operon via mutation of the repressor *frmR*. Our studies have identified 4 amino acid substitutions that inactivate FrmR: P5R, E6stp, G47S and V86D. All but E6stp are unique mutations first described here. Previous studies of *frmR* focus more on formaldehyde sensing rather than deregulation of *frmRAB* expression, potentially why our mutations have been overlooked [23,30]. The ALTAR mutations inactivate FrmR rather than uncouple regulation of transcription from formaldehyde sensing. FrmR is a small protein shown to consist of 3 α-helices and forms a tetramer [23]. The mutation E6stp deregulates *frmRAB* expression simply by the loss of FrmR in the cell. The other three mutations, however, do not cluster, each located within one of the 3 FrmR α-helices [23]. It is feasible to suggest that these mutations may impact either FrmR folding or FrmR tetramer formation, resulting in a non-functional repressor. However, this should be confirmed experimentally and was out of the scope of this current study.

The detoxification of formaldehyde by FrmAB is not the only route that bacteria can overcome exposure to this toxic compound [13]. In contrast to *E. coli,* methyltrophs generate formaldehyde at higher rates through methanol assimilation [15]. Therefore, these species have adapted to tolerate higher formaldehyde concentrations and exploit a range of alternative mechansims to overcome formaldehyde exposure. One example is EfgA, a formaldehyde sensor that may function to push bacteria into a formaldehyde induced stasis [15]. While such a concept could also be available to *E. coli*, *efgA* maintained its activity when expressed in *E. coli*, the continual long-term exposure to formaldehyde during MH use only generated chromosomal mutations that deregulated *frmAB* expression. Further studies are required to determine the rate of *frmR* mutations in vitro, and if *E. coli* can acquire the FDH[R] phenotype via any other genetic changes.

ALTAR outcomes were associated with an incidence risk ratio of 1.55 with respect to MH users suffering a breakthrough UTI [11]. The FDH[R] phenotype aids our understanding of the risks of UTI despite MH use. However, the frequency of FDH[R] isolation 4 out of 98 MH participants suggests there are other factors contributing to the increased risk for MH treatment. If it was simply that the UTI risk was driven by the acquisition of FDH[R] strains or *in situ* evolution, we would expect to see a much greater incidence of the FDH[R] phenotype. The accompanying urinalysis provides further insight into other risk factors that potentially can impact the calculated incidence risk ratio. Formaldehyde detection in ALTAR urines correlated to previous studies of methenamine treatment analysis in that μM formaldehyde concentrations were detected. However, the analysis also demonstrated a significant degree of variation amongst samples with ~ 30% of urines during the treatment phase of ALTAR having less than 0.7 μM formaldehyde, the detection limit of our assay. This is likely due to other, yet unidentified, host factors, as well as for ABX patients the variation could be attributed to participants known to switch arms after clinical assessment. Importantly, the data does show that the cohort remained compliant to their treatments during ALTAR. We recognise a limitation is that due to the logistics of trial management, this secondary analysis

was retrospective, that may impact formaldehyde stability. This may explain the discrepancy between this analysis and previous studies detecting formaldehyde in the range of 200–1300 μM.

The HPLC urinalysis adds further depth to our appreciation of the factors influencing the UTI risk with MH treatment. Our metabolite analysis showed that there was elevated peak areas in MH urines compared to ABX urines. This equated to 3 known components and 17 undefined. Not surprisingly this analysis identified formaldehyde and formate to be elevated in MH urines. However, an unexpected outcome was that glucose was also elevated. Glucose is a primary carbon source for *E. coli*. Therefore, this analysis suggests that the composition of MH urine is different from ABX urine. Changes in urine composition can both be detrimental and advantageous for bacterial growth. Thus, in combination with the variability in formaldehyde detection, the identification of the FDH^R phenotype and a significant change in urine composition, including primary carbon sources, these factors together provide a strong foundation to define the mechanism underpinning the elevated risk of breakthrough UTI in MH participants.

The FDH^R phenotype improves the rate of formaldehyde detoxification. However, is the rate of detoxification the driving force *in vivo* or can methenamine conversion overcome detoxification? Growth in artificial urine at pH 5.6 demonstrated that *E. coli frmR*^+ is sensitive to formaldehyde produced when exposed to > 0.5 mg/ml methenamine. Unfortunately, Δ*frmR* grows well in methenamine concentrations between 1 – 1.5 mg/ml (**Fig E in S1 Text**). This suggests that detoxification plays a significant role in MH efficacy and is dependent on the pharmokinetics of methenamine accumulation in urine. Interestingly, when performing MIC style growth assays, the data suggests a MIC ≤ 1 mM for FDH^S strains. However, measuring formaldehyde concentration during growth, inhibition is only effective until detoxification reduces the concentration to between 0.3 – 0.4 mM for both FDH^S and FDH^R strains (**Fig F in S1 Text**), consistent with Jordan et al (2022) [31]. These data suggests that the bacteriostatic concentration of formaldehyde is > 0.3 mM. During MH treatment such concentrations are feasible to reach with > 0.25 mg/ml of methenamine. Furthermore, previous studies have shown that for *E. coli* to maintain its population in the bladder a given growth rate is required [32]. This follows the dynamics similar to chemostat growth of bacterial strains [33,34]. Therefore, a measurable negative change in growth rate has the potential to allow the natural cycle of urine voiding to aid the removal of slow growing *E. coli* from the bladder thus reducing the opportunity for symptoms to develop.

This study generates a foundation for future studies to investigate these factors and their impact on MH use. From a clinical perspective, appreciating the rate of formaldehyde production and the maintenance of urine concentrations during treatment is key to assess how to prevent or at least reduce the selective pressure for *E. coli* to adapt and acquire FDH^R. There is also the impact of pH to consider. There is evidence that pH may impact the reactivity of formaldehyde [35]. In the context of this work, there is scope to now assess how such factors could improve MH use. It is now recognised that our urine harbours its own microflora and is not a sterile site. Therefore, using similar protocols, as used in ALTAR, further work is now necessary to appreciate the impact of MH treatment and formaldehyde detoxification on the urobiome. Indeed, consistent with our findings that urine composition is altered by MH use, Khan et al (2025) has shown in a small population of MH users (n = 10) that potential changes in the urobiome are detectable [36].

These data provide clinicians identifiable factors that can be investigated via biochemical and/ or microbiological diagnostic procedures. Detectable biomarkers improve clinical confidence during assessing the use of any treatment. The identification of both host and bacterial factors that may influence the outcome of treatment with MH will no doubt refine the use of this promising non-antibiotic preventative treatment. We recognise that these data also require clinicians and patients to re-examine their view of treatments like MH. MH has it merits to provide a means to improve antibiotic stewardship when treating recurrent UTI patients. However, a mindset change is needed by clinician and patients to accept that MH, like antibiotics, has its own associated risk of resistance. Importantly, for FDH^R specifically, it is not unforeseeable that one route of patient management would include a course of antibiotic treatment with the aim to remove FDH^R from the patient's microflora, only allowing return to MH when and if the FDH^R strains are eradicated.

In conclusion, this study has highlighted factors that may influence the efficacy of MH and may allow prediction of treatment success based on their presence or absence. This may be the first step to moving towards a more personalised treatment strategy for rUTI patients given the importance of treatment success from a patient, clinician and global perspective.

## Methods and materials

### Ethics statement

The use of samples and bacterial isolates from ALTAR participants was included in the study protocol approved by the North East Tyne and Wear South Research Ethics Committee (15/NE/0381). Formal written consent was obtained from all ALTAR participants prior to the start of their treatment phase.

### ALTAR urines and *E. coli* isolates

All ALTAR urine samples were collected by the mid-stream clean catch approach, to mimic routine heathcare practice in the UK. Participants urines (baseline, 3, 6, 9, 12, 15 and 18 months) were stored at -80°C until use [18]. *E. coli* isolated from fresh urine was taken from diagnostic CPSE plates during ALTAR and stored at -80°C with 10% DMSO when the diagnostic criteria were met. Perianal swabs were collected either under supervision or by a healthcare practitioner. Participants were made aware that they were optional. For perianal swab isolates, *E. coli* was identified using differential diagnostic media [18]. ALTAR participants that had provided ≥ 5 urine samples (n = 681) constituted 58% of all samples from 65 MH and 51 prophylactic antibiotic (ABX) participants. We have recently described the sequence analysis of 191 ALTAR derived *E. coli* isolates [19]. The genome sequence accession numbers are presented in **Table B in S1 Text** and sequence data is available: PRJEB85317 (https://www.ebi.ac.uk/ena/browser/home).

### Growth assays

Growth assays were performed using batch cultures normalised to a starting OD600 of 0.02 or in 96-well flat-bottomed microplates using protocols previously described [27,37]. Assays were performed either in Mueller Hinton Broth (MHB) or artificial urine using the protocol of Keith *et al* (2024) [38]. For our assays in artificial urine, we chose pH 6.5, 6.0 and 5.6 as they reflect the physiological pH in urine and were shown by Musher and Griffith (1974) to impact methenamine conversion. The pH of artificial urine was adjusted using 1 M HCl. Formaldehyde was diluted in MHB before addition to experimental samples. Methenamine was added to artificial urine immediately before experimental cultures were diluted. Microplates were incubated at 37°C in a BMG Fluostar Optima microplate reader with OD600 readings taken every 7 minutes with orbital shaking for 300 seconds. All batch culture experiments were performed at 37°C with constant shaking at 160 rpm.

### Formaldehyde assay

Formaldehyde was detected in urine and spent bacterial culture supernatant using Fluoral-P (4-amino-3-penten-2-one) (Sigma-Aldrich). Fluoral-P was diluted to a working concentration of 5 mM in PBS pH 6.0 from a 2 M stock resuspended in DMSO. A standard curve of formaldehyde from 0 to 3000 µM was used to calculate the formaldehyde concertation. One hundred microliters of Fluoral-P were added to 100 µl of urine or spent culture, incubated at RT for 1 hour and measured at 420 nm.

### HPLC of urine samples

A 500 µl aliquot of urine was centrifuged for 5 minutes at 10000 rpm to clear any cellular material or debris. Two-hundred microliters were transferred into a HPLC vial and loaded onto an UltiMate 3000 uHPLC system (Thermo Fisher Scientific)

using a RefractoMax521 refractive index detector and a VWD-3100 variable-wavelength detector set at A210. Ten micro-liters of sample were injected into the Aminex HPX-87H column equilibrated in 5 mM sulphuric acid. Data integration was performed manually using the Chromeleon software before further analysis in R.

### CRISPR/cas two step recombineering

The genetic manipulation protocol for *E. coli* has been described elsewhere [27,28]. All recombinants were sequenced over the region targeted to confirm the desired genetic change was present. Plasmids and strains used or constructed are described in **Table C in S1 Text**, PCR primers are declared in **Table D in S1 Text**.

### Expression analysis

RT-qPCR expression analysis was performed as described [37]. RNA was isolated using a Promega wizard kit and further treated with TurboDNase. All PCR primers used in the analysis have been described previously (**Table D in S1 Text**). Relative expression was calculated using the $2^{-\Delta\Delta Ct}$ method [39].

### Bioinformatics, data and statistical analysis

To determine the phylogeny of the ALTAR *E. coli* isolates a pangenome analysis using Panaroo v1.1.2 was performed [40]. The function panaroo-msa generated a core genome alignment and processed using FastTree2 [41]. The FastTree newick data file was imported into R and visualised with formaldehyde screening data using ggtree [42]. All other data was analysed in either Excel or R using customised scripts available on request. Statistical tests were performed using the in-built functions of excel and R.

## Supporting information

**S1 Text. Fig A: Average data of 3 independent repeats of MIC Growth assays for 10 of the 11 declared isolates identified as having an MIC > 1 mM formaldehyde in the screen shown in Fig 2.** The naming of these isolates reflects their association with MH user cases A, B, C & D: All names are in the format [Case]: [Swab (S) or Urine (U)] [Month isolated]. The 11th isolate, E: S0, is shown in **Fig B** using a restricted formaldehyde concentration range (1–2 mM). Case E: S0 was the weakest FDH^R isolate identified with respect to its MIC for formaldehyde. **Fig B:** Further average data of 3 independent repeats of MIC Growth assays for the case E isolate and controls identified in the screen shown in Fig 2. A concentration range of 1, 1·25, 1·5, 1·75 and 2 mM formaldehyde was used to demonstrate the growth advantage of Case E versus the FDH^S control and the case B: U5 as a FDH^R control isolate. **Fig C:** Schematic representation of the genetic architecture around *frmA* identified in the 11 FDH^R isolates compared to the *frmRAB* operon and its surrounding genes in CFT073. This Fig was generated using the clinker unix software package [1]. All isolates are defined by the data represented in Fig 3 and Table 1. Clinker generated schematics are aligned to *frmA* for context. **Table A:** Comparison of components that showed a significant difference in average peak area (PA). **Fig D in:** Average data of 3 independent repeats for MIC Growth assays for defined strains in artificial urine at three different pH with formaldehyde added. The data shown is the average area under curve from data sets like that shown in **Figs A and B**. Error bars are omitted for clarity. The concentrations of formaldehyde used were 0, 0.5, 075, 1, 1.5 and 1.75 mM. This data shows that all strains respond in a similar manner showing no pH dependency to increasing concentrations of formaldehyde. **Fig E:** Average AUC data of 3 independent repeats for MIC Growth assays for defined strains in artificial urine at three different pH with methenamine added at T0. Error bars are omitted for clarity. The concentrations of methenamine used were 0, 0.25, 0.5, 0.75, 1.0 and 1.5 mg/ml. This data shows methenamine conversion to formaldehyde is pH dependent only impacting E. coli growth at pH 6.0 and pH5.6. The strongest response to Methenamine was at pH 5.6 consistent with the pH dependent conversion of methenamine defined by Musher & Griffith (1974) [Ref 12 in main paper]. **Fig F:** Average growth of *frmR*^+ (CFT073)

and Δ*frmR* grown in the presence of 1 mM formaldehyde. Data represents that data used to derive the rate of detoxification shown in Fig 4C. Growth of both *frmR*⁺ and Δ*frmR* were inhibited until the formaldehyde concentration was reduced to between 0.3-0.4 mM via detoxification. **Table B:** Clinical isolates used in formaldehyde resistance screen. **Table C:** Strains and Plasmids used or constructed in this study. **Table D:** Primer used in this study.
(DOCX)

## Acknowledgments

We would like to thank the participants of ALTAR for allowing the use of samples taken during their involvement in this trial.

## Author contributions

**Conceptualization:** Frank Sargent, Judith Hall, Priyanka Krishnaswamy, Chris Harding, Phillip D. Aldridge.

**Data curation:** Niamh C. Hodgkinson, Tabarak Al-Rubaye, Thomas C. P. Reed, Catherine Mowbray, Daniel Sarkissian, Louise Cowley, Phillip D. Aldridge.

**Formal analysis:** Niamh C. Hodgkinson, Thomas C. P. Reed, Catherine Mowbray, Frank Sargent, Chris Harding, Phillip D. Aldridge.

**Funding acquisition:** Frank Sargent, Judith Hall, Priyanka Krishnaswamy, Chris Harding, Phillip D. Aldridge.

**Investigation:** Niamh C. Hodgkinson, Tabarak Al-Rubaye, Thomas C. P. Reed, Catherine Mowbray, Daniel Sarkissian, Louise Cowley, Judith Hall, Phillip D. Aldridge.

**Methodology:** Thomas C. P. Reed, Daniel Sarkissian, Louise Cowley.

**Project administration:** Phillip D. Aldridge.

**Resources:** Judith Hall, Chris Harding.

**Supervision:** Thomas C. P. Reed, Catherine Mowbray, Daniel Sarkissian, Frank Sargent, Judith Hall, Chris Harding, Phillip D. Aldridge.

**Validation:** Catherine Mowbray, Priyanka Krishnaswamy, Chris Harding, Phillip D. Aldridge.

**Visualization:** Niamh C. Hodgkinson, Catherine Mowbray, Phillip D. Aldridge.

**Writing – original draft:** Frank Sargent, Chris Harding, Phillip D. Aldridge.

**Writing – review & editing:** Niamh C. Hodgkinson, Tabarak Al-Rubaye, Thomas C. P. Reed, Catherine Mowbray, Daniel Sarkissian, Louise Cowley, Frank Sargent, Judith Hall, Priyanka Krishnaswamy, Chris Harding, Phillip D. Aldridge.

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
