## [Decision Letter · Decision Letter 0]

19 Jan 2026

Implications for methenamine hippurate use in recurrent urinary tract infection management: Formaldehyde resistance and altered urinary composition

PLOS Pathogens

Dear Dr. Aldridge,

Thank you for submitting your manuscript to PLOS Pathogens. After careful consideration, we feel that it has merit but does not fully meet PLOS Pathogens's publication criteria as it currently stands. Therefore, we invite you to submit a revised version of the manuscript that addresses the points raised during the review process.

We look forward to receiving your revised manuscript.

Kind regards,

Kimberly A. Kline

Academic Editor

PLOS Pathogens

Eva Heinz

Section Editor

Editor-in-Chief

PLOS Pathogens

PLOS Pathogens

orcid.org/0000-0002-7699-2064

**Journal Requirements:**

2) Thank you for including an Ethics Statement for your study. Please include:

i) A statement that formal consent was obtained (must state whether verbal/written) OR the reason consent was not obtained (e.g. anonymity). NOTE: If child participants, the statement must declare that formal consent was obtained from the parent/guardian.].

4) We have noticed that you have uploaded Supporting Information files, but you have not included a complete list of legends. Please add a full list of legends for your Supporting Information files after the references list.

2) If any authors received a salary from any of your funders, please state which authors and which funders..

6)  Please ensure that the funders and grant numbers match between the Financial Disclosure field and the Funding Information tab in your submission form. Note that the funders must be provided in the same order in both places as well.

7) Please send a completed 'Competing Interests' statement, including any COIs declared by your co-authors. If you have no competing interests to declare, please state "The authors have declared that no competing interests exist". Otherwise please declare all competing interests beginning with the statement "I have read the journal's policy and the authors of this manuscript have the following competing interests"

**Reviewers' Comments:**

Reviewer's Responses to Questions

**Part I - Summary**

Reviewer #1: This paper takes on a fascinating question: does an “antiseptic” lead to resistance in a treatment setting. Methenamine isn’t an antibiotic, but it is converted to the toxic molecule, formaldehyde, when the pH is sufficiently low. While I find the idea that this paper argues against to be incredibly naïve – that only antibiotics generate resistance – I am thrilled to see someone address this. They clearly show that there are formaldehyde resistant UTI strains of E. coli and that those alleles contribute to survival with MH at low pH. Overall I think what they did was fairly solid.

Reviewer #2: This submission is a secondary analysis of both urine samples and E. coli isolates obtained those samples (and from perianal swabs) during the ALTAR study., which compared symptom outcome in adult females with recurrent UTI who were treated prophylactically with either methenamine hippurate (MH) or with one of a number of antibiotics. One finding of the ALTAR study was “...a marginally higher rate of UTI episodes in those taking MH compared to the prophylactic antibiotic group. The goal of the authors was “...to understand what factors could contribute to this elevated risk of a UTI during MH use compared to daily low dose antibiotics.” The authors found that a few E. coli isolates obtained from a small subset of the participants exhibited substantially elevated minimal inhibitory concentrations for formaldehyde. Some of these formaldehyde-resistant (FDHR) isolates had inactivating mutations in the gene that encodes FrmR, the repressor protein for the frmAB operon, which encodes the formaldehyde detoxification system. Other isolates carried plasmids that encoded homologs of frmAB. This finding of elevated resistance is significant for two reasons. (1) The proposed mode of action for MH is its dissociation in the urinary tract into methenamine and hippuric acid. The hippuric acid reduces the urine pH, which leads to dissociation of the methenamine into formaldehyde and ammonia. It is the formaldehyde that is thought to inhibit bacterial growth. (2) MH is considered to be an antiseptic and thus not subject to development of resistance. The authors discovery that E. coli can evolve increased resistance (elevated MIC) requires a reevaluation of the risks associated with prophylactic MH treatment. The finding of these FHDR isolates led the authors to investigate a number of parameters associated with these isolates, MH treatment, and formaldehyde concentrations.

I consider the science to be rigorous and have absolutely no problems with it. My issues are all about the writing. The Abstract is terribly confusing. I had to read the Results to understand what the authors were trying to say in the Abstract. This is a secondary analysis, which the authors do not explicitly state. Because the Methods come after the Discussion, important details necessary for the reader to understand the Results are not provided. The text for some figures is not written in a simple intuitive fashion. The upshot is that it took me a lot longer to follow the science than it should have.

Issues with the Abstract

Lines 33 - 38. These sentences are confusing. They only became clear after I read the Results. The authors state that E. coli uses the frmRAB operon to detoxify formaldehyde and that, unexpectedly, a small percentage of isolates were able to grow in the presence of formaldehyde. They tell us that some fmrR alleles encoded non-functional variants and that others carried plasmids that encode frmA homologs. But the reader cannot understand the significance of these findings without some information concerning the roles of FrmR and FrmAB. It turns out that FrmR is the repressor for the operon. If it is non-functional, then the operon is expressed constitutively, leading to more detoxification of formaldehyde. FrmA is an enzyme involved in the detoxification. The authors must rewrite this part of the abstract, so it makes immediate sense to the reader.

Line 40. The urine composition was altered in what way? A little more detail would be helpful, especially since 2 of the alterations were formaldehyde and formate, and another was glucose. Furthermore, the authors should mention that formate is the result of formaldehyde detoxification.

Issues concerning Methods

Line 135. Since the methods are after the Discussion, the Results must explain the original ALTAR study design sufficiently for the reader to understand the Results including the following:

1) First, the authors should state explicitly that this is a secondary analysis of the original ALTAR study.

2) It is reasonable to expect the authors to tell us upfront that there were 2 arms to the study (MH and ABX), that the antibiotic treatment was prophylactic in nature, and that several different antibiotics were prescribed in the ABX arm. At least then the reader won’t be surprised to read about a participant on Nitrofurantoin on line 172.

3) The duration of treatment.

4) The duration of the sample collection.

5) That both urine samples and perianal swabs were obtained. Otherwise, lines 179-180 come as a complete surprise. See comment above.

6) The collection method of the urine, presumably the midstream voided method.

7) The number of participants in each arm and how many of those were analyzed in this secondary analysis. Don't expect the reader to count the numbers in panel B.

8) Whether there was any significant difference in any demographic or symptomatic variable between the 2 arms. I don't expect a table of the participants, but it is important to know that the ALTAR study did or did not find any significant differences between the participants in the 2 arms.

Line 159. This comes as a complete surprise. What isolates? How were they isolated? They were sequenced? I'm not asking for the methods here. I'm asking that you state that you isolated E. coli somehow and that you sequenced their genomes. Then tell us that you screened them.

Issues with figures & tables.

I like Figure 1. It is well designed and quite intuitive. Ditto for Figure 2.

Figure 3. This figure is either not intuitive or not well described. I get what the authors are trying to say but it took me a while to understand all of the parts of the figure. More explanation would be helpful in the text and in the legend.

More Figure 3. What do the circles and squares mean? And on line 199, the authors say “see legend.” This is the legend. You mean the information in panel A. I would say “see Panel A.”

Line 218. I think the phrase "order of cases" is confusing. I think you mean that they refer to the cases shown in Figure 3. It's the word "order" that is problematic.

Figure 4A. Why is this figure oriented the opposite of the previous growth curves?

Line 317: Figure 6. Same problem as above. "See legend" is confusing because this is the legend. Maybe the best way is to say something like "See upper left panel."

Figure S1. The legend should say that these are 10 of the 11 resistant isolates so the reader understands the statement about isolate E: S0.

Fig. S1. The legend says that Case E: S0 is shown in Figure S3. It is actually S2.

Figure S3 legend. What is "the clinker?" Is it a bioinformatic tool? The problem is the word "the." Delete it in this sentence and in the last one.

Also, the reference 8 is not in the bibliography.

Issues concerning dogma

Lines 33, 110, 157. Concerning the use of the phrases “uropathogen E. coli” and “major uropathogen E. coli.” I don't want to pontificate but there are 2 issues with these phrases. First, there are no obvious differences between the supposed UPEC strains and their closest relatives. What appears to be more important is the phylogroup of the isolate. Second, there’s good reason to support Arturo Casadevall’s contention that the more appropriate term is “bacterium with pathogenic potential.” Pathogenesis requires a host; therefore, what does it mean to call a bacterium a pathogen? The way I’ve been handling these issues is to use the term pathogenic potential and to call CFT073, UTI89 and NU14 classic UPEC strains.

Line 76: This is old dogma. It assumes that dysbiosis is not a cause. One doesn't need to say "invading." Delete it.

Comments concerning the Discussion

The authors appear to be unaware of a relatively recent paper that shows that methenamine treatment does not necessarily eliminate pathogens (Khan et al., 2025 PMID: 41317299). Instead, it alters urinary microbiome composition. This is likely related to the authors observation that treatment with MH alters the composition of the urine. The authors might want to read the Khan paper and comment on it in their Discussion. It’s relevant.

**Part II – Major Issues: Key Experiments Required for Acceptance**

Please use this section to detail the key new experiments or modifications of existing experiments that should be absolutely required to validate study conclusions.required to validate study conclusions.

Reviewer #1: Very naïve discussion of formaldehyde. This is probably the biggest weakness. Throughout there are many omissions and some errors. There is actually a pretty decent literature base in the roles of aldehydes in pathogens. This has even led to an “aldehyde hypothesis” that cells generate these to help kill pathogens. I have pasted in a few references below that can serve as a starting place for this:

Hazen SL, Hsu FF, d’Avignon A, Heinecke JW. 1998 Human neutrophils employ myeloperoxidase to

convert alpha-amino acids to a battery of reactive aldehydes: a pathway for aldehyde generation at sites of

inflammation. Biochem 37:6864–6873.

Singh S, Brocker C, Koppaka V, Chen Y, Jackson BC, Matsumoto A, Thompson DC, Vasiliou V. 2013.

Aldehyde dehydrogenases in cellular responses to oxidative/electrophilic stress. Free Radic Biol Med

56:89-101. doi: 10.1016/j.freeradbiomed.2012.11.010, PMC3631350

Chen NH, Djoko KY, Veyrier FJ, McEwan AG. 2016. Formaldehyde stress responses in bacterial

pathogens. Front Microbiol 7:257. doi: 10.3389/fmicb.2016.00257, PMC4776306

Zhao J, Missihoun TD, Bartels D. 2017. The role of Arabidopsis aldehyde dehydrogenase genes in

response to high temperature and stress combinations. J Exp Bot 68:4295-4308. doi: 10.1093/jxb/erx194,

PMC5853279

Tsai HY, Hsu YJ, Lu CY, Tsai MC, Hung WC, Chen PC, Wang JC, Hsu LA, Yeh YH, Chu P, Tsai SH.

2021. Pharmacological activation of aldehyde dehydrogenase 2 protects against heatstroke-induced acute

lung injury by modulating oxidative stress and endothelial dysfunction. Front Immunol 12:740562. doi:

10.3389/fimmu.2021.740562, PMC8576434

Darwin KH, Stanley SA. 2022. The aldehyde hypothesis: metabolic intermediates as antimicrobial

effectors. Open Biol 12:220010. doi: 10.1098/rsob.220010, PMC9006002

Second, there is a fair amount more about E. coli and formaldehyde out there. Some of it is directly related (see the heterogeneity in resistance coming from higher frmA in the first paper) and other points to distinct mechanisms to explore (thioproline in the 2nd). In the first paper, they find an identical behavior: the timing of E. coli recovery was set by having to reduce the formaldehyde added to below a threshold.

https://www.biorxiv.org/content/10.1101/2022.09.23.509177v1

https://www.biorxiv.org/content/10.1101/2020.03.19.981027v1.full

Third, evolution to increased formaldehyde has been done in several organisms. The results are not straight-forward. Putting in “evolution of formaldehyde resistance” into google pulls up a paper (below) using a methanol eating bug, Methylobacterium, where the mutations were found to be in novel sensors and NOT in formaldehyde oxidation. The major point here is that you -could- have found mechanisms other than the known formaldehyde oxidation pathway as the beneficial mutation in your evolved strains, but you didn’t.

https://journals.plos.org/plosbiology/article?id=10.1371/journal.pbio.3001208

In discussing the E. coli pathway, FrmA is not an oxygenase. No molecular oxygen is involved. It is an S-hydroxymethyl-GSH dehydrogenase that is NAD-dependent (generating NADH). It is also important to mention that formaldehyde reacts with reduced glutathione (often stated as GSH) and not with the oxidized form (written GSSG). When the formyl group is hydrolyzed, it is recycled as GSH (thus no net reduction is needed to regenerate the cofactor).

Another point of concern is the effect of pH on the reactivity of formaldehyde. Low pH is needed to break down methenamine. There is, however, a second effect. At low and high pH formaldehyde is also less reactive with molecules such as the nitrogen groups of pterins. I have seen this demonstrated in the following paper. Because of this, E. coli is known to reduce the cellular pH when aldehydes (like methylglyoxal) build-up by using the Kef efflux pumps (many papers from Ian Booth’s lab).

• 10.1128/JB.182.23.6645-6650.2000

It is great to see a lab phenotype for the frmR deletion strain. However, this was only a single hour of lag. And the patients were treated for a year. Is that a big enough difference? I think that needs some substantial discussion, for it isn’t clear to me that there would be very strong selection then. Most likely, the lab is not the same as a patient, of course, but this needs to be addressed.

Reviewer #2: I require no new experiments.

**Part III – Minor Issues: Editorial and Data Presentation Modifications**

Reviewer #1: p.5, line 112 – Here and elsewhere, the S in S-hydroxymethylglutathione (and other similar compounds) should be italicized.

p.7, line 132 – The urine concentration is stated as ~1 millimolar and then the numbers listed below are 0.88 to 122.1 micromolar. Are both of these units correct?

p.11, line 191 – Correct “overtime” to “over time”

p.12, line 210 – If the loci on the plasmids are also frmRAB, as I believe is being stated, then they would be called redundant (homologous extra copies) rather than degenerate (non-homologous, alternative pathways to accomplish the same task).

p.12, line 211 – For the substitutions, it is hard to read them as FrmRE6stp. I would change all of these to just E6stp, etc., since they all occur in the same protein (FrmR). They are much easier to understand that style in table 1.

p.13, line 226 – Relevant to the point made a couple points above, rather than how similar the adhC is to a plasmid, it would be most helpful to know how similar they are to the chromosomal frmRAB cluster. Might even want to turn to AA identity or similarity if they are very different.

Reviewer #2: Other comments:

Line 36. The proper word is "unexpectedly" not “unexpectantly.”

Line 39. The authors should tell the reader that CFT073 is a classic UPEC strain.

Line 42. Alleles do not grow. Isolates that encode alleles grow.

Line 96. Define NICE.

Line 97. The authors define the acronym MH on line 101 but it should be defined on line 97 when "methenamine hippurate" is first mentioned.

Line 121: Rate is a function of time, which the next sentence clarifies. But, this must read UTUI episodes per year.

Line 122. The authors c/should add that this could be statistically and biologically significant - i.e. development of resistance.

Lines 138-140. This statement should be made in the Discussion, not the Results. It is opinion/inference/hypothesis not results.

Line 142. "These data...." Data are plural.

Line 160. The reader does not know why you were surprised. Why were you surprised? Use your answer to write a transition phrase. Something like "Because X, we were surprised...."

Line 161. Would be helpful to state what these elevated MIC were.

Line 225. This sentence is ambiguous. Lack of sequence data of which - pVU3695 or the ALTAR plasmids? It only becomes clear when you read the next sentence.

Line 227. Again, this is ambiguous until you read the end of the sentence. Be explicit.

Line 230. But you don't have the full sequence of the plasmid. Is this 88% to adhC (frmA)? Be explicit.

Line 259. "Compliment" means to praise. The proper term is "complement." Sorry English can be a pain sometimes.

Line 296. This whole section is confusing. It’s the writing not the data. It took me a while to figure out what the authors were trying to say. The show 2 different types of data for 2 different strains (WT and mutant) at 3 different pHs. And they jump immediately to the outlier, which is - of course - the interesting information. The sentence on line 299 seems odd since the authors just said that they measured formaldehyde concentration. It would be better if they added the information that they measured growth in the previous sentence. Then it makes sense to talk about inhibition of growth. But I recommend that the authors mention the results of the higher pHs first, then talk about what happens at the lower pH.

Line 305. This sentence is not intuitive. Do you mean that WT growth was inhibited as long as the FDH remained above 0.4 mM? And that there was lag until the concentration remained above that concentration?

Lines 330, 340. Remind the readers that FrmR is a represssor.

Lines 426. Please describe the collection methods for both urines and swabs. After mentioning the urines and swabs at the beginning of the Results, the details should be included here in the Methods.

Line 427. From "from fresh urine" correct?

Issue of simple curiosity

Line 118. Interesting result. If the methenamine and hippuric acid dissociate in the kidneys, then what is the mode of action in the gut?

PLOS authors have the option to publish the peer review history of their article (what does this mean? ). If published, this will include your full peer review and any attached files.). If published, this will include your full peer review and any attached files.

**Do you want your identity to be public for this peer review?** For information about this choice, including consent withdrawal, please see our For information about this choice, including consent withdrawal, please see our Privacy Policy ..

Reviewer #1: No

Reviewer #2: **Yes:** Alan J. WolfeAlan J. Wolfe

**Figure resubmission:**

**Reproducibility:**



---

## [Decision Letter · Decision Letter 1]

14 Mar 2026

Dear Dr Aldridge,

We are pleased to inform you that your manuscript 'Implications for methenamine hippurate use in recurrent urinary tract infection management: Formaldehyde resistance and altered urinary composition' has been provisionally accepted for publication in PLOS Pathogens.

Best regards,

Kimberly A. Kline

Academic Editor

PLOS Pathogens

Eva Heinz

Section Editor

PLOS Pathogens

Sumita Bhaduri-McIntosh

Editor-in-Chief

PLOS Pathogens

orcid.org/0000-0003-2946-9497

Michael Malim

Editor-in-Chief

PLOS Pathogens

orcid.org/0000-0002-7699-2064

Consider figure font size suggestion during the copy editing step.

Reviewer Comments (if any, and for reference):

Reviewer's Responses to Questions

**Part I - Summary**

Reviewer #1: Covered this during the first stage of review.

Reviewer #2: In my original review, I said that the topic is timely and the study innovative. I considered the science to be rigorous and had no problems with it. My original issues were all about the writing, and authors have responded positively to all my concerns and those of the other reviewers. I am satisfied.

**Part II – Major Issues: Key Experiments Required for Acceptance**

Please use this section to detail the key new experiments or modifications of existing experiments that should be absolutely required to validate study conclusions.required to validate study conclusions.

Reviewer #1: No major issues remain, as the critical needs to include more work on E. coli with formaldehyde and formaldehyde resistance more broadly are now addressed.

Reviewer #2: None

**Part III – Minor Issues: Editorial and Data Presentation Modifications**

Reviewer #1: My one remaining concern is the figures. At the size they would likely be published, many of the legends and axis labels are completely invisible. Figure 4A and 6 are two examples of this. While even 4B and 4C push the edge of visible size, 4A is impossible to read. Similarly, the axes are marginally visible for figure 6, but the legend is nowhere close. Easy fixes, but definitely necessary.

Reviewer #2: None that the copy editor won't find.

PLOS authors have the option to publish the peer review history of their article (what does this mean? ). If published, this will include your full peer review and any attached files.). If published, this will include your full peer review and any attached files.

**Do you want your identity to be public for this peer review?** For information about this choice, including consent withdrawal, please see our For information about this choice, including consent withdrawal, please see our Privacy Policy ..

Reviewer #1: No

Reviewer #2: **Yes:** Alan J WolfeAlan J Wolfe

---

## [Editor Report · Acceptance letter]

Dear Dr Aldridge,

We are delighted to inform you that your manuscript, "Implications for methenamine hippurate use in recurrent urinary tract infection management: Formaldehyde resistance and altered urinary composition," has been formally accepted for publication in PLOS Pathogens.

Best regards,

Sumita Bhaduri-McIntosh

Editor-in-Chief

PLOS Pathogens

orcid.org/0000-0003-2946-9497

Michael Malim

Editor-in-Chief

PLOS Pathogens

orcid.org/0000-0002-7699-2064